# The Role of Brown Algae as a Capping Agent in the Synthesis of ZnO Nanoparticles to Enhance the Antibacterial Activities of Cotton Fabrics

**DOI:** 10.3390/md23020071

**Published:** 2025-02-07

**Authors:** Eli Rohaeti, Rasamimanana Joronavalona, Paulina Taba, Dewi Sondari, Azlan Kamari

**Affiliations:** 1Department of Chemistry Education, Faculty of Mathematics and Natural Science, Universitas Negeri Yogyakarta, Yogyakarta 55281, Indonesia; 2Department of Chemistry, Faculty of Mathematics and Natural Science, Universitas Indonesia, Depok 16424, Indonesia; helmiyati@sci.ui.ac.id; 3Department of Organic Chemistry, University of Antananarivo, VS 21 GAI Ankatso, Antananarivo 101, Madagascar; joronavalonachemist@gmail.com; 4Department of Chemistry, University of Hasanudin, Makasar 90245, Indonesia; paulinataba@unhas.ac.id; 5The National Research and Innovation Agency, Jakarta Pusat 10340, Indonesia; sondaridewi@gmail.com; 6Department of Chemistry, Universiti Pendidikan Sultan Idris, Tanjong Malim 35900, Malaysia; azlan.kamari@fsmt.upsi.edu.my

**Keywords:** brown algae extract, cotton fabric, green synthesis, silane compounds, ZnO nanoparticles

## Abstract

Research was conducted on the role of brown algae as a capping agent in the synthesis of ZnO nanoparticles, the characteristics of ZnO nanoparticles, and the effect of the addition of ZnO nanoparticles and/or silane compounds on antibacterial and antifungal activities. The synthesis of ZnO nanoparticles involved green synthesis, and then nanoparticles were characterized using UV/VIS/NIR, ATR-FTIR, XRD, PSA, and SEM-EDS, followed by the in situ deposition of ZnO nanoparticles on cotton fabrics and the addition of silane compounds. The characterization of modified and unmodified cotton fabrics and antibacterial and antifungal activity tests were carried out using the disc diffusion method through measurements of the diameter of the inhibition zone against *Pseudomonas aeruginosa*, *Staphylococcus epidermidis*, and *Malassezia furfur*. The characterization of ZnO nanoparticles showed absorption at a wavelength of 357 nm; the number of waves was 450 cm^−1^; the diffraction peak occurred at an angle of 36.14°; the crystal size was 15.35 nm; there was a heterogeneous particle distribution; the particle size was in the ranges of 1.74–706 nm (PSA) and 45–297 nm (SEM); and an irregular particle shape was noted. The results showed that the best antibacterial and antifungal activity was obtained in cotton + HDTMS + ZnO nanoparticles (K8) and cotton + ZnO nanoparticles+HDTMS/MTMS (K4).

## 1. Introduction

Cotton, spandex, polyester, and nylon are a few textile materials that can be made into goods with antibacterial and antifouling qualities [1]. Cotton is one of the natural fibers derived from plants; its main constituent is cellulose and it is used as a raw material for the textile industry. Cotton fabric has the property of easily absorbing water molecules (hygroscopic) and is biodegradable [2]. The hygroscopic properties of cotton fabric can facilitate the absorption of sweat, meaning that cotton fabric easily becomes dirty and overrun by microorganisms. Therefore, cotton fabrics can be modified to increase their resistance to microorganism and antifouling attacks by using zinc oxide (ZnO) nanoparticles and silane compounds.

ZnO nanoparticles have the ability to inhibit microbial activity by damaging cell membranes and producing ROS (Reactive Oxygen Species), which results in oxidative stress in microbes [3,4]. The antimicrobial properties of ZnO nanoparticles can protect cotton fabrics from the growth of microorganisms that can reduce the quality of the fabric, cause the colors to fade, etc. In addition to their antimicrobial properties, ZnO nanoparticles have several advantages, namely being biocompatible and photocatalytic and inhibiting UV rays. ZnO nanoparticles have a white color that does not change the appearance of the fabric, and they are cheaper to produce compared to other nanoparticles, such as silver nanoparticles. ZnO nanoparticles can be produced using two approaches to nanoparticle synthesis, namely top-down (physics) and bottom-up (chemical) approaches.

The top-down approach is carried out by breaking down large particles into small particles of a nanometer size through methods such as the laser beam irradiation method [5] and ball milling [6]. The bottom-up approach involves chemical reactions in which atoms form larger particles of nanometer size. The bottom-up approach can be achieved in several ways, such as coprecipitation, sol–gel, and chemical reduction methods. However, both approaches have weaknesses. The top-down method requires considerable energy use and has high costs. Meanwhile, the bottom-up approach uses several chemical reducing materials that are toxic and also requires considerable costs [7]. Therefore, a method of synthesizing ZnO nanoparticles was developed that is low cost and environmentally friendly, using a bottom-up approach, namely green synthesis.

The green synthesis of ZnO nanoparticles can be carried out with the help of ultrasonication [8] and microwave [9], utilizing secondary metabolite compounds from plant extracts as bio-reductors. The use of microwave and ultrasound can significantly reduce the time needed to synthesize nanoparticles, help control the particle size of nanoparticles, reduce the cost of nanoparticle synthesis by reducing the time and energy used, and also reduce the environmental impact by reducing the use of chemicals and energy. The ZnO nanoparticles in this study are the result of their preparation using brown algae (*Padina* sp.) which is derived from natural materials that are environmentally friendly. The carbohydrate content and secondary metabolite compounds of *Padina* sp., which occur in the form of flavonoids, tannins, saponins, alkaloids, and other metabolites, have -OH functional groups as bio-reducers and capping agents that can prevent particle agglomeration during the synthesis of ZnO nanoparticles [10,11].

ZnO nanoparticles that have antimicrobial properties can be deposited into cotton fabrics in situ (one step) and ex situ (two step). ZnO nanoparticles in the textile industry are generally processed using the ex situ method, in which the synthesis of ZnO nanoparticles is carried out first and then the ZnO nanoparticles are characterized and deposited onto the fabric. This method has the advantage of allowing for large-scale production; in addition, the morphology, size, and distribution of the produced ZnO nanoparticles can be controlled. However, the disadvantage of this method is that it is a complicated and inefficient process. In contrast, in situ methods can produce ZnO nanoparticles that have an uncontrollable morphology, size, and distribution. This method has high efficiency because the creation of ZnO nanoparticles occurs directly on the surface of the fabric. The stronger bond between the fabric fibers and the ZnO nanoparticles provides the fabric with resistance to different washing frequencies [12]. Therefore, this study uses an in situ method for the deposition of ZnO nanoparticles on cotton fabrics.

The cotton fabrics on which ZnO nanoparticles were deposited were characterized using UV/VIS/NIR, ATR-FTIR, XRD, and SEM-EDS instruments. The ZnO nanoparticles that were synthesized independently were characterized using UV/VIS/NIR, ATR-FTIR, XRD, PSA, and SEM-EDS instruments.

The functionalization of antifouling properties in cotton fabrics can be obtained through the addition of silane compounds that cause increased hydrophobicity. Octtiltrietoxilane (OTES), hexadecyltrimethoxylan (HDTMS), and methyltrimethoxylane (MTMS) compounds [13,14,15] show potential in the modification of textile fabrics. The silane compounds applied to cotton fabrics are HDTMS, MTMS, and HDTMS/MTMS composites. The analysis of antimicrobial properties was determined with the disc diffusion method using *Staphylococcus epidermidis* as a Gram-positive bacterium, *Pseudomonas aeruginosa* as a Gram-negative bacteriaum, and the fungus *Malassezia furfur*.

## 2. Results and Discussions

### 2.1. Green Synthesis of ZnO Nanoparticles with Brown Algae (Padina sp.)

The green synthesis process in this study used a ratio of the volume of brown algae: early precursor (Zn(NO_3_)_2_) of 0.01 M:NaOH 0.1 M, respectively, which is 1:20:5; the ratio was an optimum comparison of the experimental results. The synthesis process began with the addition of algae extract and an initial precursor (Zn(NO_3_)_2_) of 0.01 M to produce a slightly cloudy brown solution caused by brown algae extract. Synthesis reactions require heating treatment and controlled time for the reaction to be stable. The green synthesis process can be reviewed before and after the addition of NaOH by marking the presence of white deposits of ZnO nanoparticles. Then, pH measurements can be taken on *Padina* sp., solution (Zn(NO_3_)_2_) 0.01 M, and brown algae extract + solution (Zn(NO_3_)_2_) 0.01 M (before adding NaOH) and after adding NaOH. The pH values obtained consecutively were 7.58, 5.35, 6.33, and 10.69 (around 10–11). The addition of NaOH can cause an increase in pH, which has a role as a settling agent in the deposition reaction of Zn^2+^ ions in the initial precursor (Zn(NO_3_)_2_) to Zn(OH)_2_ and then to ZnO nanoparticles.

The green synthesis process of ZnO nanoparticles involves the use of natural material extracts, namely *Padina* sp. as a bio reducer and capping agent, with the possibility of a reaction to reduce Zn^2+^ ions to Zn+ to Zn^0^ and then nucleation, growth, and autooxidation into ZnO nanoparticles [16]. The green synthesis process that occurred in this study began with the interaction of biomolecular compounds of brown algae extract with Zn^2+^ ions from the initial precursor (Zn(NO_3_)_2_). After the addition of NaOH, a deposition reaction occurs, which is characterized by the formation of metal hydroxides and biomolecular compounds that play a role in binding to the Zn^2+^ ion cluster, which causes the particle surface to be enveloped by negative ions, which results in a repulsive force between similar charges so as to prevent particle aggregation [17,18]. *Padina* sp. is a type of seaweed used in the synthesis of ZnO nanoparticles for several reasons: *Padina* sp. is an abundant and easily obtained biomass, so it can be used as a source of raw material for the synthesis of ZnO nanoparticles. *Padina* sp. contains active compounds such as alginic acid, fucoidan, and other compounds that can function as reducing agents and stabilizers in the synthesis of ZnO nanoparticles. *Padina* sp. is a biocompatible biomass, so it can be used in biomedical and environmental applications. *Padina* sp. extract can function as a reducing agent to convert Zn^2+^ ions into ZnO nanoparticles. *Padina* sp. extract can also function as a stabilizer to prevent the aggregation of ZnO nanoparticles. The use of *Padina* sp. in the synthesis of ZnO nanoparticles is environmentally friendly because it does not involve the use of hazardous chemicals and can reduce waste. Thus, *Padina* sp. can be used as an environmentally friendly and biocompatible alternative for the synthesis of ZnO nanoparticles.

The Zn^2+^ ion reacts with the hydroxide ions from NaOH to form a milky white suspension mixture. White deposits can be obtained and separated due to the properties of ZnO nanoparticles that are insoluble in water. However, when in the suspension mixture, the compound formed is Zn(OH)_2_. Meanwhile, the excess OH^−^ ions will react with Zn(OH)_2_ to form a [Zn(OH)_4_]^2−^ complex. For the H_2_O molecules present in the mixture and the energy obtained when the solution is stirred using a magnetic stirrer, the [Zn(OH)_4_]^2−^ complex is dissociated again to form Zn^2+^ and OH^−^ ions, which further form solids (powders) of ZnO nanoparticles during the heating process (drying), marked by the release of water vapor [19]. Furthermore, the calcination process of ZnO nanoparticles can improve the crystallinity [20]. The green synthesis process can be reviewed in the reaction scheme shown in Figure 1 [10,17,19,21]. However, the literature shows that a biological pathway for the synthesis of zinc oxide nanoparticles using *Pseudomonas aeruginosa* rhamnolipids has already been reported [22].

### 2.2. Modification of Cotton Fabric with ZnO Nanoparticles and Silane Compounds

The curing process was carried out on cotton fabrics that were modified with ZnO nanoparticles to bind the material or ZnO nanoparticles to the surface of the fabric to make it stronger and more durable [23]. The results of the modification of ZnO nanoparticles and/or silane compounds did not show a significant qualitative difference because the color of the white cotton fabric was the same as the color of the ZnO nanoparticle deposits and silane compounds, which had clear visuals. After modification, the modified cotton fabric was characterized and tested for contact angle, mechanical properties, and antimicrobial activity.

### 2.3. Analysis of ZnO Nanoparticle Function Cluster and Cotton Fabric With and Without Modification

Based on the results of UV/VIS/NIR tests in the wavelength range of 200–800 nm, the spectra of ZnO nanoparticles, cotton fabrics, and cotton fabrics + ZnO nanoparticles showed absorption peaks at wavelengths of 327 nm and 357 nm, respectively, showing no absorption peaks, and 329 nm. The peak absorption of ZnO nanoparticles is in the wavelength range of 300–380 nm [10,22]. The peak absorption results have different values in the research that has been carried out, namely 320 nm [24], 327 nm [25], 355 nm [26], 357 nm [27], and 365 nm [8].

The peak uptake of ZnO nanoparticles varies due to the influence of diverse particle sizes. Particles that are not necessarily uniform certainly have absorption peaks at different wavelengths. The decreasing particle size causes a decrease in absorption and a blue shift (a shift in wavelength from long to short). In addition, there is a decrease in the UV A/UV B ratio and the effect of quantum size on absorbance with changes in particle size [28].

The difference in absorption peaks in cotton fabrics and cotton fabrics deposited with ZnO nanoparticles is found at the wavelength of 329 nm, which is estimated to be the wavelength of ZnO nanoparticles. The presence of absorption peaks at these wavelengths indicates that ZnO nanoparticles have been deposited on the cotton fabrics. The value of one of the peaks of absorption of ZnO nanoparticles (327 nm) is different from that of the ZnO nanoparticles deposited in cotton fabrics (329 nm) due to the presence of cotton fabrics, which can cause the signal captured by the detector to be less noisy, although the difference is not too significant.

Tests that were carried out with the ATR-FTIR spectrophotometer instrument produced spectral data that show the interpretation of the functional cluster. The ATR-FTIR K0 spectra show the peak absorption position at 3323, 3257, 2884, 1637, 1425, 1363, 1312, 1015, and 898 cm^−1^. Meanwhile, the ATR-FTIR K1 spectra in Figure 2 show the peak absorption positions of 3323, 3250, 2886, 1636, 1423, 1382, 1312, 1021, 900, and 452 cm^−1^. Wide absorption bands in regions 3323, 3257, and 3250 cm^−1^ with an area range of 3500–3200 cm^−1^ indicate the presence of vibration stretching of the O-H group from cellulose (cotton) [29].

The absorption band positions in the 2884 and 2886 cm^−1^ regions showed the presence of vibration stretching of the C-H group derived from polysaccharides in cellulose and brown algae extract compounds used as capping agents in the process of depositing ZnO nanoparticles in situ [30]. The absorption bands in the region of 1637 and 1636 cm^−1^ are characterized as water molecules found in the cotton fabric (cellulose). Small absorption bands in the 1425 and 1423 cm^−1^ regions indicate the presence of the CH_2_ symmetric bending of cellulose. The position of the absorption peak in the regions (1353 and 1382 cm^−1^) and 1312 cm^−1^ respectively shows the presence of bending vibration of the C-H and C-O groups. Wide absorption peaks in the 1015 and 1021 cm^−1^ regions showed a vibration relationship between C-O and O-H stretching in polysaccharides from cellulose. In addition, the peak of uptake in the regions of 898 and 900 cm^−1^ is characterized as a β-glycosidic relationship with monosaccharides [31].

The ATR-FTIR spectra show the peak absorption positions of 2362, 2338, 908, 669, and 450 cm^−1^. The peak of absorption in the region of 2362 cm^−1^ indicates the presence of a C=O functional group [32]. The position of the absorption band in the area of 2338 cm^−1^ indicates the vibration of the stretching of the Amina group (C=N) [33]. In addition, the absorption band in the area of 908 cm^−1^ shows the presence of C-H out-of-plane bending [34]. The band of absorption in the region of 669 cm^−1^ is characterized as a Zn-OH functional group [35]. The position of the absorption band in the region of 450 cm^−1^ is characterized as a Zn-O functional group that enters the wave number range of the Zn-O functional group, which is 650–400 cm^−1^ [33,36].

The K0 and K1 spectra have almost the same spectrum, but in K1, there is a differentiating absorption peak at 452 cm^−1^, which is characterized as the Zn-O functional group. This value indicates that the deposition of ZnO nanoparticles on cotton fabrics has been successfully carried out. The comparison of the spectral range of K0, K1, and ZnO nanoparticles can be seen in Figure 2.

The ZnO nanoparticles formed have a Zn(OH)_2_ component with the identification of the Zn-OH functional group, which indicates that the drying process needs to be studied further so as to produce ZnO nanoparticles that are close to high purity. The peak of uptake found in the spectrum of ZnO nanoparticles indicates the presence of an analyzed organic functional group. This is in accordance with the role of biomolecular compounds contained in brown algae extract as a capping agent. The biomolecular compounds that can be predicted, namely alkaloids, are indicated by the identification of C=O, C-H, and C=N functional groups. Biomolecular compounds that play a definite role in green synthesis need to be further studied using compound elucidation methods, such as GC-MS, 13C-NMR, and 1H-NMR analysis.

### 2.4. Analysis of the Diffraction Patterns and Crystal Size of ZnO Nanoparticles

Tests that were carried out with XRD spectrometer instruments produced diffraction pattern data so that the crystal size could be clearly determined. The test was carried out on ZnO nanoparticles and cotton fabrics deposited with ZnO (K1) nanoparticles using an angle range of 2θ in the range of 3–90° and 4–80°, respectively. The results of XRD analysis were processed using the X’pert HighScore Plus program to match with the XRD database, namely the Crystallography Open Database (COD), and the OriginLab version 2023b program to determine the FWHM value that can be used in determining crystal size with the Debye–Scherrer equation shown in Equation (1) [9]. The diffraction pattern of ZnO nanoparticles and cotton fabrics deposited with ZnO (K1) nanoparticles can be reviewed in Figure 3 and Figure 4.(1)$$D=\frac{k\lambda }{\beta cos\theta }$$

With:

*D* = Crystal size (nm)

*k* = Form factor constant (0.91 for hexagonal shapes)

*λ* = Wavelength (Cu-Kα1 = 0.15406 nm)

*β* = FWHM (*Full Width at Half Maximum*) (radian)

*θ* = X-ray diffraction angle (radian)

Based on Figure 3, the diffraction pattern of ZnO nanoparticles that has been confirmed with COD data 96-900-4180 shows matched peaks at 2θ = 31.63°; 34.3°; 36.14°; 47.42°; 56.52°; 62.76°; 66.28°; 67.87°; 68,92°; 72,54°; 76,84°; and 89.47°, with Miller index values (h k l) of (1 0 0), (0 0 2), (1 0 1), (1 0 2), (1 0 2), (1 1 0), (1 0 3), (2 0 0), (1 1 2), (2 0 1), (0 0 4), (2 0 2), and (2 0 3), respectively. Based on matching with the COD database, the crystal structure of ZnO nanoparticles from green synthesis using *Padina* sp. forms the hexagonal crystal structure of wurtzite (zincite). In addition, the crystal size of the ZnO nanoparticles from the Debye–Scherrer equation is 15.35 nm at the peak position of the crystal with a Miller index value (1 0 1). The resulting peaks have values that do not differ significantly from the COD database. Some unmatched peaks indicate the presence of impurities from compounds derived from the algae extract of *Padina* sp., meaning that there can be a decrease in the crystallinity of ZnO nanoparticles. This can occur due to the influence of temperature during the calcination process. Successive increases or decreases in calcination temperature can increase or decrease the crystallinity and crystal size of ZnO nanoparticles [37].

Meanwhile, the diffraction pattern of cotton fabric deposited with ZnO nanoparticles, shown in Figure 4, shows peaks at 2θ = 14.37°; 16.15°; 22.40°; 25.96°; and 34.11°, identified as crystalline cellulose. The peaks cannot be identified with the COD database due to the number of phases being more than one. In addition, the ZnO nanoparticles in cotton fabrics are shown at a peak of 35.98°. The peaks are not crystalline (amorphous) and shift somewhat because the identified phases are not dominant compared to the peaks that show crystalline cellulose. The crystal size of ZnO nanoparticles in cotton fabrics can be determined using the Debye–Scherrer equation assuming the phase in which crystals are formed so that a crystal size of 20.43 nm is obtained. 

### 2.5. Results of Particle Size and Distribution Analysis of ZnO Nanoparticles

Analysis of particle size and distribution using the Particle Size Analyzer (PSA) instrument was carried out. The analysis began with the dispersion of ZnO nanoparticles with a concentration of 0.1% (b/v) using ethanol and then sonication for 30 min. The results of the PSA data of ZnO nanoparticles are reviewed in Table 1.

Based on Table 1, PSA data were obtained, which include a PDI (Polydispersity Index) of 0.537, a nanoscale particle size in the range of 1.74–706 nm, and a microscale particle size in the range of 3860–5710 nm. The polydispersity index value is in the range of 0–1, which indicates the particle size distribution. A PDI value close to zero indicates a homogeneous particle distribution, while a polydispersity index value of >0.5 indicates that the particles have a high level of heterogeneity [38]. The results of the PSA data showed that the particles were still not homogeneously distributed, which could be reviewed at a PDI value of >0.5, and the size of the ZnO particles was in the nano- and micro-regions. The dominant particle size is on the nanoscale (1–1000 nm) with a total percentage of 91.2%. However, the ZnO particles that are still present at the microscale (3860 and 5710 nm) have a percentage of 4.1 and 4.7%, indicating that the particles formed are agglomerating. This is suspected to be due to the effectiveness of *Padina* sp. as the amount of capping agent is still low. Conversely, a strong capping agent role can limit the growth of ZnO clusters and prevent aggregations from being large. The formation of nanoparticles can be affected by the pH of the reaction solution, the concentration of seaweed, the concentration of metal salts (precursors), the time and temperature of the reaction, and the temperature and time of drying and calcining [37,39].

The difference occurred between the results of particle measurements using PSA instruments and crystal measurements using the Debye–Scherrer equation. This is because the Debye–Scherrer equation used is an estimate of the size of the crystal material and not the size of the particle. One particle is made up of a number of tiny crystals. A particle that is nanometer in size by having one crystal in one particle indicates that the crystal size is equal to the particle size [39]. On the other hand, the results of measuring ZnO particles using *Padina* sp. extract in this study with the PSA instrument showed particles that were nanometers and micrometers in size, so the crystals obtained became heterogeneous and did not necessarily indicate the particle size.

### 2.6. Analysis of yjr Morphology, Particle Form, and Elemental Composition of ZnO Nanoparticles

The results obtained were in the form of SEM images that could inform on the shape and morphology of the surface of ZnO nanoparticles, cotton fabrics (K0), cotton fabrics deposited with ZnO nanoparticles (K1), and cotton fabrics deposited with ZnO nanoparticles and silane compounds (HDTMS/MTMS) (K4). In addition, the elemental composition of each sample tested was determined using EDS analysis. Testing was carried out with several SEM instruments (Thermo Fisher Scientific Axia ChemiSEM, Phenom Pro X G6, and JEOL JSM-6510LA) and FESEM (JEOL JSM-IT700HR) with coating using a sputter coater that uses conductive target materials such as Au and Au/Pd to increase the conductivity of the sample in minimizing the electron charging effect, which can have an impact on the quality of SEM image resolution. The results of SEM and EDS images of ZnO, K0, K1, and K4 nanoparticles with varying magnifications can be reviewed in Figure 5, Figure 6, Figure 7 and Figure 8.

SEM image results with a magnification of 50,000× using JEOL JSM-6510LA at an accelerating voltage (AV) of 10 kV are shown in Figure 5.

Particle measurements using SEM can be performed directly using the tool or image processing software such as ImageJ. Particle measurement with a magnification of 50,000×, as shown in Figure 5, yields a particle size in the range of 45–297 nm. Meanwhile, at smaller magnifications such as 10,000× and 20,000×, particles with varying sizes from the nanometer and micrometer scales show an uneven distribution of particles. In addition, the 50,000× magnification shown in Figure 5 shows a heterogeneous particle distribution. This is due to the ZnO particles that have undergone agglomeration events, meaning that the results of particle size and distribution analysis using SEM are in accordance with the PSA results.

Analysis of particle shape can be observed at high magnifications such as 100,000× and 200,000× showing irregularly shaped particles such as rod-like and leaf-like particles. The irregular shape of the particles can be caused by several factors such as precursor concentration, pH conditions, heating temperature during the reaction, stirring speed, reaction time, and other factors. Therefore, it is necessary to further study the factors that affect the irregularity of particle shape.

Figure 6 shows the EDS results of ZnO nanoparticles at a magnification of 20,000×. The EDS results for ZnO nanoparticles can be analyzed qualitatively in the presence of several identified elements, namely C, O, and Zn. The mass percentage (%) of the identified elements was 6.6, 17.9, and 75.5, respectively. Meanwhile, the known atomic percentages (%) were 19.4, 39.6, and 41, respectively. The presence of element C in ZnO nanoparticles indicates that there is still a capping agent derived from biomolecular compounds.

### 2.7. SEM Images and Composition of Cotton Fabrics (K0)

Morphological analysis of cotton fabrics was performed using SEM Phenom Pro X G6 with magnifications of 1000×, 2000×, and 5000×, which can be seen in Figure 7. The shape of the cotton fabric observed in the SEM images is in accordance with the theory, namely, cotton fibers in the form of ribbons or tubes that collapse and twist. The surface of the raw cotton fiber has a clean area and very few contaminants can be observed. This was confirmed using the EDS analysis, with it only containing elements C and O derived from the constituent of cotton fabrics, namely cellulose. The results of the EDS analysis can be reviewed in Figure 8, which shows the mass and atomic percentage. The mass percentage (%) identified was 47.6 and 52.4, respectively. Meanwhile, the atomic percentage (%) was 54.8 and 45.2, respectively.

### 2.8. SEM Images and Composition of Cotton Fabrics After Modification with ZnO (K1) Nanoparticles

The morphological analysis of cotton fabrics deposited with ZnO nanoparticles using Axia ChemiSEM at magnifications of 1000× can be seen in Figure 9. The surface of cotton fabrics that have been deposited with ZnO nanoparticles looks rougher than cotton fabrics without modification. The presence of ZnO nanoparticles deposited on cotton fabrics can be confirmed by the EDS results and those shown in Figure 9.

The results of EDS analysis (Figure 10) at a magnification of 1000× showed the presence of elements C, O, and Zn. The identified mass percentages (%) were 23.7, 41.3, and 35, respectively. Meanwhile, the atomic percentages (%) were 38.8, 50.7, and 10.5, respectively. The presence of the Zn element indicates that the deposition of ZnO nanoparticles on cotton fabrics using the in situ method has been successfully carried out. Element C in K1 shows the presence of cellulose as a constituent of cotton fabrics and biomolecular compounds that act as capping agents in in situ deposits of ZnO nanoparticles. Based on the mass percentage from smallest to largest, namely, elements C, Zn, and O, respectively, the sum of the mass percentage of the element Zn is smaller than O because the Zn of the deposited ZnO nanoparticles is unevenly deposited and is less dominant than cotton fabrics composed of elements C and O.

### 2.9. SEM Images and Composition of Modified Cotton Fabrics with ZnO Nanoparticles and HDTMS/MTMS

Morphological analysis of cotton fabrics deposited with ZnO nanoparticles and silane compounds (HDTMS/MTMS) (K4) using Axia ChemiSEM at a magnification of 1000× can be seen in Figure 11.

Based on the SEM image in Figure 11, the surface of K4 is characterized by the presence of ZnO nanoparticles that have an uneven distribution but that are more homogeneous or uniform with particles in the form of micro-nanoflowers. This can be seen in the SEM images with a magnification of 17,500×.

ZnO deposited on cotton fabric has a smaller number of agglomerated particles compared to K1. The growth of ZnO clusters in K4 is easier to control than in K1 due to the addition of silane compounds as coupling agents by increasing the adhesion force and interface interaction of ZnO nanoparticles or matrix polymers such as cotton fabrics. Coupling agents have the ability to balance the hydrophilic and hydrophobic properties of a polymer or membrane material, thereby increasing the hydrophobic properties of the material. Increased hydrophobic properties can reduce the water absorption in ZnO nanoparticles so that agglomeration events in ZnO nanoparticles can be minimized [40].

ZnO particles in the form of micro-nanoflowers can be measured with the help of ImageJ software at a magnification of 17,500×. Particle sizes are obtained in the range of 224–1688 nm. Higher magnification can make it easier to measure ZnO nanoparticles, but the resolution of the SEM image is reduced, meaning that the resulting image becomes more opaque. In addition, the nanoparticles of ZnO and silane compounds (HDTMS/MTMS) deposited on cotton fabrics can be confirmed by the results of EDS analysis at the magnification of 17,500×, as shown in Figure 12.

The EDS results at 17,500× magnification using AV of 15 kV showed the presence of elements C, O, Zn, Si, and Au. The mass percentage (%) identified were 61.7, 32.9, 4.5, 0.7, and 0.2, respectively. Meanwhile, the atomic percentages (%) were 45.9, 32.5, 18, 1.2, and 2.4, respectively. The presence of the element Zn indicates that the deposition of ZnO nanoparticles on cotton fabrics has been successfully carried out. Element C in K1 shows the presence of cellulose as the dominant constituent of cotton fabrics and biomolecular compounds that act as capping agents in ZnO nanoparticle deposits. In addition, the element Si shows the presence of silane compounds as a coupling agent that can increase the hydrophobicity of cotton fabrics. Meanwhile, the analyzed Au element comes from the Au target material in increasing the conductivity of cotton fabric samples. The peaks of Zn and Au in the energy range of 8–10 keV respectively show the X-ray characteristics at Kα and Lα energies. This is due to the use of AV of 15 kV so that the peaks can be analyzed.

### 2.10. Antibacterial Activity of Cotton Fabric With and Without Modification

Antibacterial activity can be seen from the clear zone that forms or what is known as the inhibition zone around the cotton fabric sample. The formed resistance zone can be measured to find out the magnitude of the diameter through three sides, namely horizontally, vertically, and diagonally. The effectiveness of antibacterial activity can be observed through the formation of inhibitory zones. Bacterial growth inhibitory responses are classified into four groups, namely weak (≤5 mm in diameter), medium (5–10 mm in diameter), strong (10–20 mm in diameter), and very strong (≥20 mm in diameter) [41].

Antibacterial testing was carried out on cotton fabrics without modification as well as after modification (K0–K10). In addition, tests were also carried out on ZnO nanoparticles dispersed in aquabidest with a concentration of 1000 ppm (Z). The use of a positive control (K+) such as ciprofloxacin results in an inhibitory zone with a large diameter against Gram-positive (*Staphylococcus epidermidis*) and Gram-negative (*Pseudomonas aeruginosa*) test bacteria [42]. Meanwhile, the negative control (K−) used is aquabidest. Positive and negative controls were used as a comparison of sample test results. Antibacterial testing was carried out for 48 h with the measurement time being every 3 h.

### 2.11. Measurement of the Inhibition Zone Against Pseudomonas aeruginosa

Antibacterial testing on modified cotton fabrics was carried out by measuring the inhibition zone against *Pseudomonas aeruginosa*. The results of the sample inhibition zone measurements compared to the positive and negative controls can be reviewed in Figure 13.

Based on Figure 13 showing the results for the antibacterial test samples, the results are almost the same, so some samples look like they overlap, except for samples K8 and Z, which showed maximum values at the 42nd hour of 3.93 mm and 2.79 mm, respectively. Based on the graph, the formation of a large inhibition zone diameter in the test sample tends to be at the 42nd hour. After the 42nd hour, the test sample experienced resistance to the test bacteria, indicated by the instability events up and down the diameter of the inhibition zone of the test sample. However, before the 42nd hour, the test sample also had values that tended to go up and down. This can be caused by the medium used, the environmental conditions during testing such as temperature and humidity, the stability of antibacterial substances, the condition of cotton fabric samples, the incubation time, metabolic activity, and the number of bacteria.

The K8 modified cotton fabric sample type has the highest value when compared to the other modifications and the one without modification (K0). This shows that the K8 sample type has the highest antibacterial activity against the bacterium *Pseudomonas aeruginosa*. HDTMS-modified ZnO nanoparticles provide a larger surface area for bacterial interaction. In addition, the hydrophobicity of HDTMS helps maintain the stability of ZnO nanoparticles, ensuring prolonged antibacterial activity [43,44,45]. Meanwhile, ZnO nanoparticles have the second-highest antibacterial activity. The incubation time from the 0th hour to the 9th hour does not show an inhibition zone, so it begins with the 9th hour. In addition, the best incubation time is at the 42nd hour as evidenced by the results of measuring the diameter of the maximum inhibition zone.

### 2.12. Measurement of Inhibition Zones Against Staphylococcus epidermidis

Measurements of the inhibitory zone on modified cotton fabrics against *Staphylococcus epidermidis* can be reviewed in Figure 14. In addition, measurements were also taken on unmodified cotton fabrics (K0), ZnO nanoparticles (Z), and positive (K+) and negative (K−) controls.

Inhibition zone measurements were also taken three times per sample and Figure 14 shows that one of the samples has a large inhibition zone. The results of measuring the diameter of the inhibitory zone of *Staphylococcus epidermidis* in Z, K0–K10, and K+ samples are presented. In addition, K− was also measured, but an inhibition zone was not formed.

Based on the results of the ANOVA test, a post hoc ANOVA follow-up test was then carried out, namely, the Tukey HSD (Honestly Significant Difference), test to determine the variation in sample type and the best incubation time in the antibacterial test against *Staphylococcus epidermidis.* Based on the results of the Tukey HSD antibacterial test against *Staphylococcus epidermidis* bacteria, it was shown that the K8 modified cotton fabric sample type had the highest value when compared to the other modifications and that without modification (K0). This shows that the K8 sample type has the highest antibacterial activity against *Staphylococcus epidermidis*. HDTMS and ZnO nanoparticles have a synergistic effect. HDTMS and ZnO nanoparticles can synergistically enhance the formation of ROS, thereby enhancing antibacterial efficacy. HDTMS-modified ZnO nanoparticles provide a larger surface area for bacterial interaction. In addition, the hydrophobicity of HDTMS helps maintain the stability of ZnO nanoparticles, ensuring prolonged antibacterial activity [44,45,46]. Meanwhile, the K9 sample type showed almost the same results as K8 so it did not differ significantly. ZnO nanoparticles had the third-highest value, which proves the existence of antibacterial activity against *Staphylococcus epidermidis*. In addition, based on the results of the Tukey HSD test, the best incubation time was at the 48th hour followed by the results of measuring the maximum diameter of the inhibition zone. The incubation time of the 6th, 15th, 30th, 27th, 33rd, 39th, and 42nd hours showed results close to the 48th hour, resulting in almost the same antibacterial activity.

### 2.13. Antifungal Activity of Cotton Fabric With and Without Modification

Measurement of antifungal activity inhibition zones that have similarities with the working principle of antibacterial activity measurement. Antifungal testing was carried out on cotton fabrics without modification as well as after modification (K0–K10). In addition, tests were also carried out on ZnO nanoparticles dispersed in aqua bidest with a concentration of 1000 ppm (Z). The use of a positive control (K+) such as ketoconazole results in an inhibitory zone with a large diameter against the fungus *Malassezia furfur* [46]. Meanwhile, the negative control (K−) used was aquabidest. Positive and negative controls are used as a comparison of sample test results. Antifungal testing was carried out for 48 h with a measurement time of every 6 h. The measurement of the inhibition zone for antifungal activity can be reviewed in Figure 15.

Based on Figure 15, the antifungal test samples showed almost the same results, so some samples showed overlapping and stagnant increases, except for samples K4 and K8, which showed maximum values of 16.47 mm and 16.78 mm at the 42nd and 48th hours, respectively. The stagnant state can be affected by the medium used, the environmental conditions during testing such as temperature and humidity, the stability of antifungal substances, the condition of cotton fabric samples, the incubation time, metabolic activity, and the number of fungi.

The positive control (K+) can be included in the graph because the value of the inhibition zone formed does not show much difference from the K4 and K8 samples. The maximum resistance zone that can be formed at K+ is 14.07 mm at the 42nd hour measurement. The value of the maximum inhibition zone of the positive control is smaller than that of the test sample due to the improper quality, particle size, and concentration of K+, so it is necessary to further study the particle size and the minimum inhibition concentration of the positive control. The decrease in the value of the inhibition zone occurs at the 48th hour, which indicates that the medium has the potential to rise or decrease after 48 h, so the antifungal measurement is better than 48 h in order to find out the best inhibition zone.

Based on the results of the Tukey HSD antifungal test against *Malassezia furfur* fungus, they show that the K4 modified cotton fabric sample type has the highest value when compared to other modifications and that without modification (K0). This shows that the K4 sample type has the highest antifungal activity against *Malassezia furfur*. Meanwhile, K4 is the type of sample that has the second-largest inhibition zone value, showing almost the same results statistically as K8, so it does not differ significantly. The Tukey HSD test was then carried out to determine the variation in incubation time, showing the best incubation time, which was at the 42nd hour followed by the 48th hour. This is in accordance with the results of the measurement of the diameter of the maximum resistance zone that occurred in K4 and K8, respectively. Simplification of the Tukey HSD antifungal test against *Malassezia furfur* fungus on sample type variations can be reviewed in Table 2.

The fungal growth inhibitory response of Z, K+, and K0–K10 samples, except K4 and K8, was classified in the weak group. Meanwhile, the K4 and K8 samples could be classified into the strong and medium groups, respectively, because the cotton cloth had been modified with ZnO nanoparticles and/or silane compounds that provided interactions between compounds in terms of antifungal activity in the fungus *Malassezia furfur*. Therefore, further research is needed on the effects of the interaction between ZnO nanoparticles and/or silane compounds on antifungal activity in the *Malassezia furfur* fungus. ZnO nanoparticles, HDTMS, and MTMS synergistically enhance ROS formation in cotton composites, attacking *Malassezia furfur*. Silane composites enhance the dispersion of ZnO nanoparticles [47]. HDTMS and MTMS can maintain the stability of ZnO nanoparticles. Thus, ZnO nanoparticles, HDTMS, and MTMS composites can cause long-lasting antifungal activity in cotton fabrics.

Mechanism of antibacterial and antifungal properties [44,45,46,47] include the following: 1. Surface area: Smaller NPs (10–100 nm) have larger surface areas, increasing interactions with microorganisms; 2. Reactive Oxygen Species: NPs generate ROS, damaging microbial cell membranes and DNA; 3. Cell membrane disruption: NPs interact with bacterial cell membranes, causing structural changes and leakage; 4. DNA binding: NPs bind to microbial DNA, inhibiting replication; and 5. Protein denaturation: NPs disrupt essential protein functions.

However, the size, distribution, and shape of nanoparticles can affect antibacterial activities. Smaller NPs (<10 nm) can enhance penetration into microbial cells, increasing antibacterial efficacy. Larger NPs (10–100 nm) can cause a greater surface area, improving ROS generation and cell membrane disruption. Optimal size (20–50 nm) can balance penetration, ROS generation, and surface area for maximum antibacterial activity [44,45,46]. Nanoparticle distribution affects antibacterial activities. Uniform distribution can ensure consistent antibacterial activity across fabric surfaces. Aggregation can reduce efficacy due to decreased surface area and ROS generation. Cluster formation can enhance antibacterial activity through localized ROS generation [44,45,46,47]. Nanoparticle shape can cause antibacterial activities. Spherical NPs are the optimal shape for ROS generation and cell membrane disruption. Rod-shaped NPs can increase surface area, improving antibacterial activity. Triangular/Hexagonal NPs can enhance ROS generation due to increased surface roughness [44,45,46].

## 3. Materials and Methods

The instruments used in this study were a UV/VIS/NIR spectrophotometer (Shimadzu UV-3600 Plus, Shimadzu Corporation, Kyoto, Japan), an ATR-FTIR spectrophotometer (Shimadzu IRSpirit QATR-S, Shimadzu Corporation, Kyoto, Japan), an XRD spectrometer (Rigaku MiniFlex 600 and Shimadzu LabX XRD-6000, Rigaku Corporation, The Woodlands, TX, USA), a PSA (Microtrac Nanotrac Wave II, Shimadzu, Kyoto, Japan), an SEM-EDS (Axia ChemiSEM, Phenom Pro X G6, Thermo Fisher Scientific, Waltham, MA, USA, and JEOL JSM-6510LA, Jeol Ltd., Tokyo, Japan), a FESEM (JEOL JSM-IT700HR, Phenom Scientific, Eindhoven, The Netherlands), a sputter coater (cressington Sputter Coater 108auto/SE (Cressington Scientific Instruments Ltd., Watford, UK), Quorum SC7620 (Quorum Technologies Ltd., Lewes, UK), JEOL JEC-3000FC and Smart Coater DII-29030SCTR from Jeol USA Headquarters and Technology Center, New England, ME, USA), an ultrasonic cleaner (GT Sonic, Guangdong GT Ultrasonic Limited Company, Shenzhen, China), laminar air flow (LAF, Thermo Fisher Scientific, Waltham, USA), glassware, a micropipette, a magnetic stirrer, a blender, an 80-mesh test sieve, an oven, a muffle furnace, centrifuges, vacuum filters, pH meters, refrigerators, shakers, analytical scales, HDPE plastic bottles, thermometers, statives, clamps, hair dryers, cooler boxes, a digital capillary, hole punches, autoclaves, incubators, Petri dishes, an Ose wire, and a drigalski.

The materials were cotton fabric and brown algae (*Padina* sp.) taken from Drini Beach, Gunung Kidul, Special Region of Yogyakarta, Indonesia. The brown color of the algae is caused by the pigment fucoxanthin, which belongs to the xanthophyll group and is more dominant than other pigments such as chlorophyll. Zn(NO_3_)_2_·6H_2_O was purchased from Sigma Aldrich (MilliporeSigma, Darmstadt, Germany, Methyltrimethoxylane (MTMS > 98% from Sigma Aldrich, MilliporeSigma, Darmstadt, Germany), hexadecyltrimethoxyxylane (HDTMS > 85% from Sigma Aldrich, MilliporeSigma, Darmstadt, Germany), ethanol 96% (Merck Millipore, MilliporeSigma, Darmstadt, Germany), NaOH, aquabidest, acetone, *Staphylococcus epidermidis* (FNCC 0048 strain), *Pseudomonas aeruginosa* (FNCC 0063 strain), *Malassezia furfur*, Nutrient Agar (NA) (Oxoid, Thermo Fisher Scientific, Shrewsbury, UK), Nutrient Broth (NB) (Merck Millipore, MilliporeSigma, Darmstadt, Germany), Potato Dextrose Agar (PDA) (Oxoid, Thermo Fisher Scientific, Shrewsbury, UK), and Whatmann filter paper no. 41 were used in this study.

### 3.1. Extraction of Brown Algae (Padina sp.)

The brown algae was washed using running water and then dried in an oven for 24 h at 50 °C until it had a water content of around 5%, and then it was cooled for 30 min at room temperature. After that, the dried brown algae was ground into powder form, so that it had a long shelf life and to make extraction easier. A total of 1.5 g of *Padina* sp. powder with a size of 100 mesh was dissolved in 150 mL of distilled water and heated at 100 °C for 25 min, with occasional stirring so that the solution was not saturated. After it was heated, the mixture was cooled and centrifuged for 30 min at a speed of 5000 rpm until the supernatant and pellet were produced. The separated supernatant in the form of brown algae extract was filtered using Whatman filter paper no. 41 with the help of a filter vacuum so that the extract results obtained were more optimal. The resulting filtrate extracted was put into an HDPE plastic bottle and stored in a refrigerator with a temperature of 4 °C.

### 3.2. Preparation of 0.1 M NaOH Solution

A 0.1 M NaOH solution was prepared by dissolving 1 g of solid NaOH in a 250 mL distilled water and homogenizing it.

### 3.3. Green Synthesis of ZnO Nanoparticles

Green synthesis begins by mixing 25 mL of brown algae extract (with aquabidest solvent) and 500 mL of 0.01 M zinc nitrate solution (Zn(NO_3_)_2_). The mixture was then stirred and heated at 80 °C for 5 min and then cooled at room temperature for 30 min. Then, 125 mL of NaOH solution was added slowly until the pH of the mixture was around 10–11, before it was placed in a refrigerator for 24 h at 4 °C. After that, the mixture was centrifuged at a speed of 5000 rpm for 30 min until the supernatant and pellets were produced. The pellets were separately washed 3 times to minimize dirt aquabidest. The pellets were dried in an oven at 100 °C for 6 h, and then we continued with the calcination process using a muffle furnace at a temperature of 400 °C for 4 h until the pellets became ZnO nanoparticles powder.

### 3.4. ZnO Nanoparticle Characterization

ZnO nanoparticle characterization tests were carried out using UV/VIS/NIR, ATR-FTIR, XRD, PSA, and SEM-EDS instruments.

In Situ Deposition of ZnO Nanoparticles onto Cotton Fabric (K0) was cut to a size 15 × 15 cm. It was then washed using acetone solution with soaking for 30 min and then soaked using distilled water for 30 min. Next, the cloth was dried using a hair dryer. Cotton cloth (K0) or modified cotton cloth (K5, K6, and K7) was placed in a beaker containing 5 mL of brown algae extract (Padina sp.) and 100 mL of 0.01 M zinc nitrate (Zn(NO_3_)_2_) solution. It was then stirred and heated at 80 °C for 5 min and then cooled at room temperature for 30 min. We then added 25 mL of NaOH solution slowly until the pH of the mixture was around 10–11. The mixture was transferred to a 250 mL Erlenmeyer flask and then shaken with a shaker for 24 h. The fabric that had been deposited with ZnO nanoparticles in situ was dried in an oven for 30 min at a temperature of 100 °C and then cured for 5 min at a temperature of 140 °C to obtain cotton fabric that had been deposited with ZnO nanoparticles (K1 or K8, K9, and K10).

### 3.5. Modification of Cotton Fabric with Silane Compound

Surface modification of unmodified cotton fabric (K0) or modified cotton fabric (K1) with silane compounds was carried out by immersing the fabric in an Erlenmeyer flask containing 300 mL of HDTMS in a solution of 20 mL and 10 mL of ethanol and aquabidest, respectively, for 4 h. Next, the fabric was removed and dried in an oven at 100 °C for 30 min to obtain samples K2 or K5.

Surface modification of unmodified cotton fabric (K0) or modified cotton fabric (K1) with silane compounds was carried out by immersing the fabric in an Erlenmeyer flask containing 300 μL of MTMS in a solution of 20 mL and 10 mL of ethanol and aquabidest, respectively, for 4 h. Next, the fabric was removed and dried in an oven at 100 °C for 30 min to obtain samples K3 or K6.

Surface modification to cotton fabric (K0) or modified cotton fabric with silane compounds (K1) was carried out by immersing the fabric in an Erlenmeyer flask containing a mixture of 100 μL and 150 μL of HDTMS and MTMS in 20 mL ethanol and in 10 mL distilled water, respectively, for 1.5 h. Next, the fabric was removed and dried in an oven at 100 °C for 30 min to obtain samples K4 or K7.

### 3.6. Antimicrobial Activity Assay of Cotton Fabric

The antimicrobial activity test began with the process of sterilizing the equipment and media using an autoclave. The media were made from nutrient agar (NA) for bacteria and potato dextrose agar (PDA) for fungi. Next, the bacteria *Pseudomonas aeruginosa* and *Staphylococcus epidermidis* and the fungus *Malassezia furfur* were placed in one cycle of nutrient broth and then incubated with a shaker incubator at a speed of 124 rpm for 24 h. Then, the sterilized NA or PDA was put into a Petri dish, about ½ of the Petri dish covered by it, and then left for 24 h until the media became solid.

The solid media were inoculated with the bacteria or fungus using a micropipette and leveled using a drigalski. After that, antimicrobial activity testing was carried out using the disc diffusion method by analyzing the inhibition zone formed around the paper disc or disc sample which is marked by the width of the clear zone. This test was carried out using a paper disc and a sample shaped like a disc with a diameter of 6 mm. Bacterial inhibition zones were measured using digital calipers every 3 h, while for the fungus, this took place every 6 h. The formation of a clear zone indicates antimicrobial activity capabilities.

## 4. Conclusions

The success of ZnO nanoparticle synthesis using *Padina* sp. as a capping agent is indicated by the absorption at a wavelength of 357 nm; the Zn-O functional group in the wave number region of 450 cm^−1^; the peak of diffraction being at an angle of 36.14° with a Miller index of (1 0 1), indicating a crystal size of 15.35 nm; the heterogeneous and uneven particle distribution; the particle size in the range of 1.74–706 nm (PSA) and 45–297 nm (SEM); and the irregular particle shapes such as rod-like and leaf-like. There was a significant difference in antibacterial activity between modified and unmodified cotton fabrics against *Pseudomonas aeruginosa* and *Staphylococcus epidermidis*. Cotton fabrics with the addition of HDTMS compounds and modification of ZnO (K8) nanoparticles showed the best results in inhibiting bacterial growth with the best time variation of each bacterium at the 42nd and 48th hours, respectively. There was a significant difference in antifungal activity between modified and unmodified cotton fabrics against *Malassezia furfur.* Cotton fabrics with ZnO nanoparticle modification and the addition of HDTMS and MTMS composites (K4) showed the best results in inhibiting fungal growth with the best time variation at 42 h.

## Figures and Tables

**Figure 1 marinedrugs-23-00071-f001:**
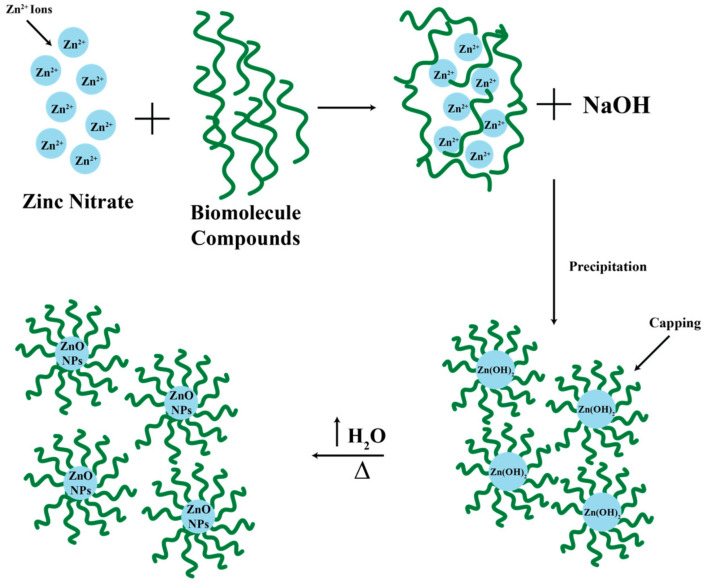
Green synthesis reaction scheme of ZnO nanoparticles.

**Figure 2 marinedrugs-23-00071-f002:**
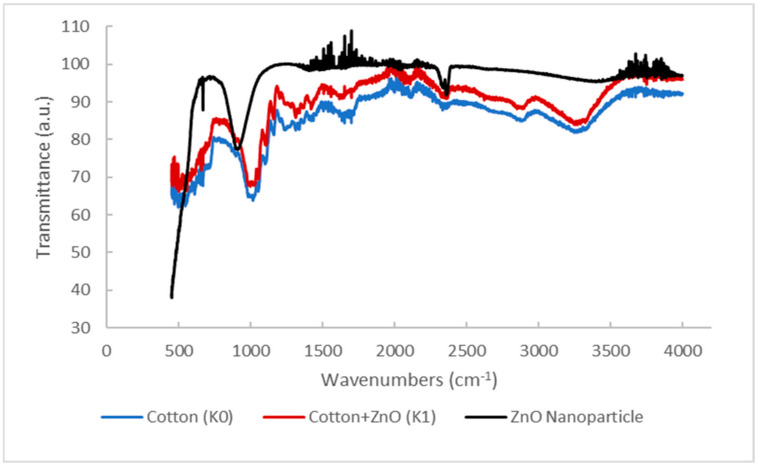
Comparison of the FTIR spectra of K0, K1, and ZnO nanoparticles.

**Figure 3 marinedrugs-23-00071-f003:**
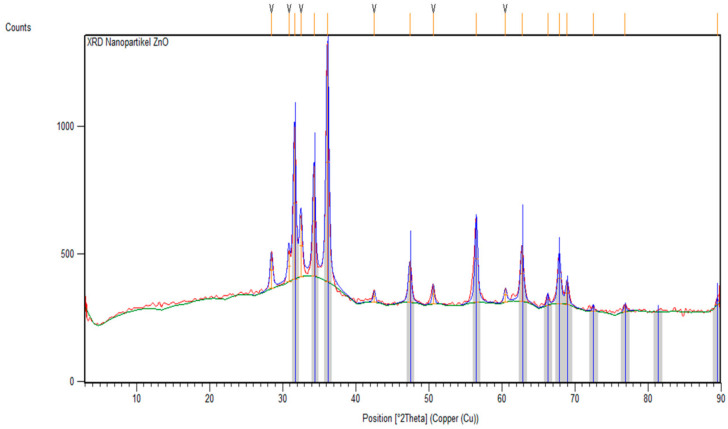
XRD diffractogram of ZnO nanoparticles compared with the COD Database 96-900-4180.

**Figure 4 marinedrugs-23-00071-f004:**
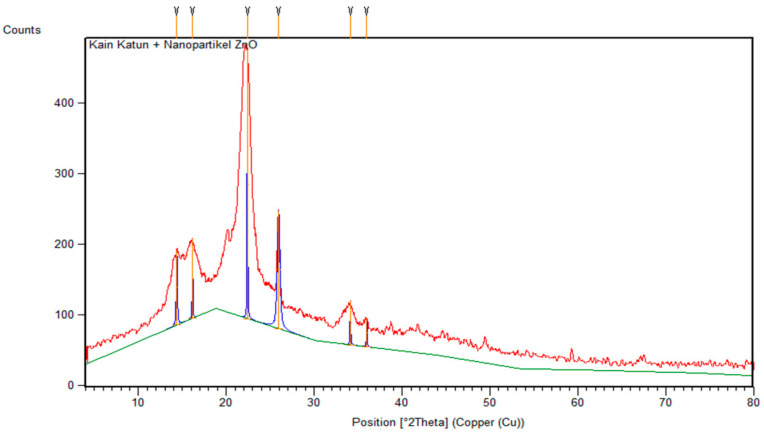
XRD diffractogram of cotton fabric-deposited nanoparticles ZnO (K1).

**Figure 5 marinedrugs-23-00071-f005:**
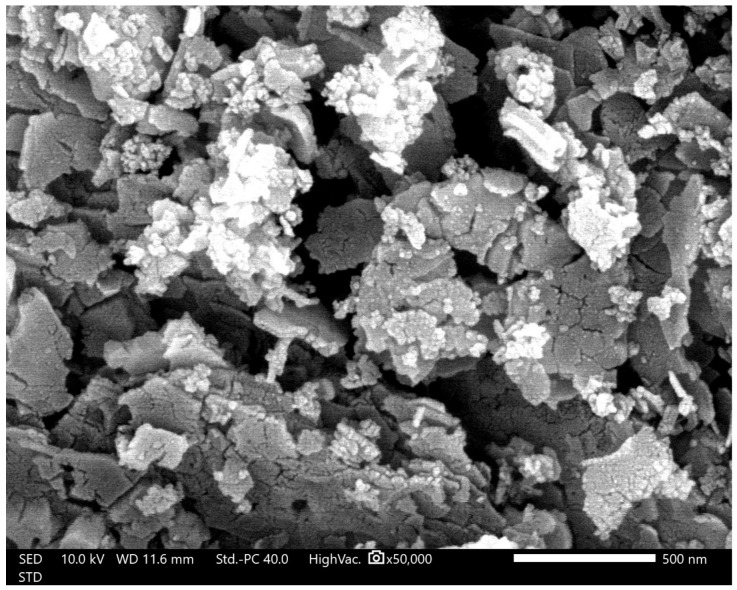
SEM image of ZnO nanoparticles with JEOL JSM-6510LA. Magnification: 50,000×.

**Figure 6 marinedrugs-23-00071-f006:**
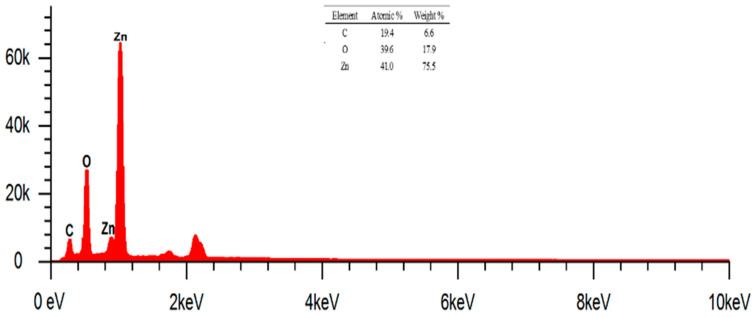
EDS results of ZnO nanoparticles.

**Figure 7 marinedrugs-23-00071-f007:**
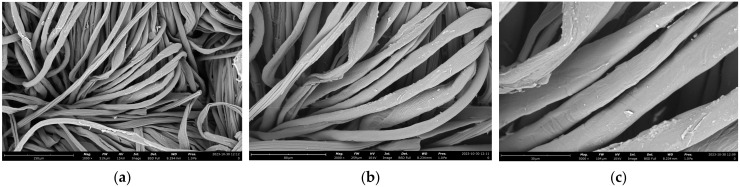
SEM image of cotton fabric with Phenom Pro X G6. Magnification: (**a**) 1000×; (**b**) 2000×; (**c**) 5000×.

**Figure 8 marinedrugs-23-00071-f008:**
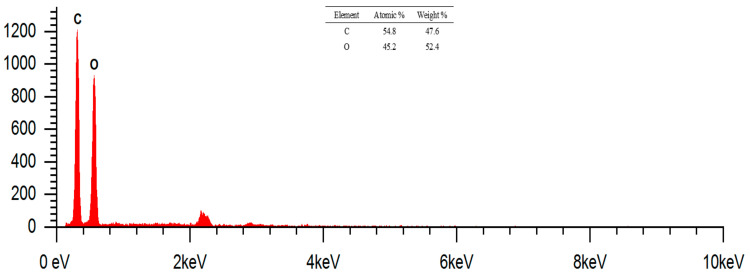
EDS region results for cotton fabric. Magnification: 5000×.

**Figure 9 marinedrugs-23-00071-f009:**
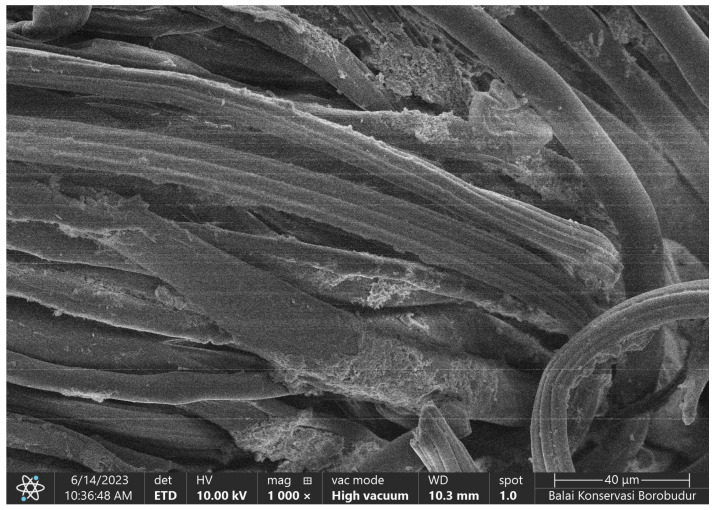
SEM image of cotton fabric + ZnO nanoparticles (K1) with Axia ChemiSEM. Magnification: 1000×.

**Figure 10 marinedrugs-23-00071-f010:**
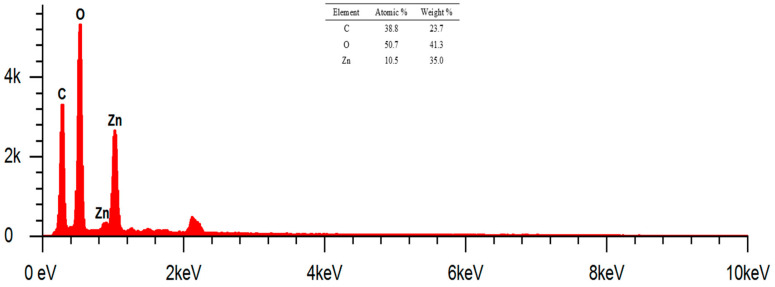
EDS results of cotton fabric + nanoparticles ZnO (K1). Magnification: 1000×.

**Figure 11 marinedrugs-23-00071-f011:**
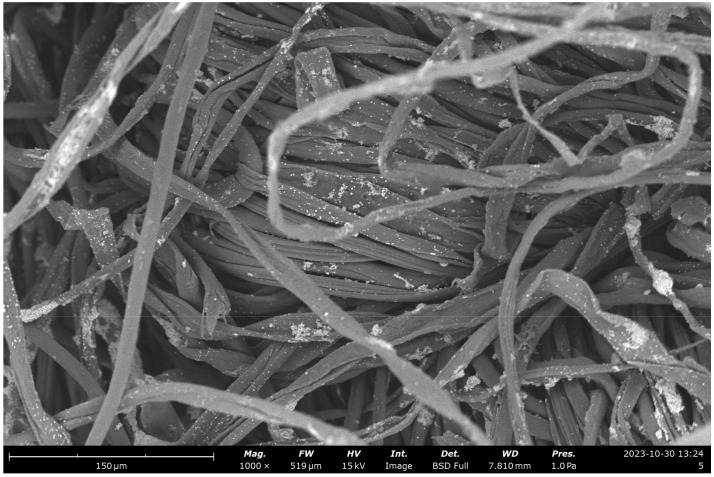
SEM image of cotton fabric + nanoparticles ZnO + HDTMS/MTMS (K4) with Phenom ProX G6. Magnification: 1000×.

**Figure 12 marinedrugs-23-00071-f012:**
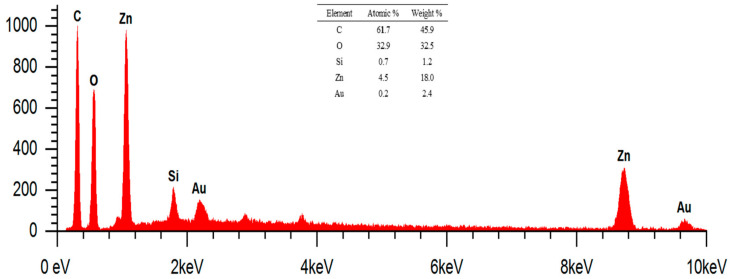
EDS results for spot cotton fabric + nanoparticles ZnO + HDTMS/MTMS (K4). Magnification: 17,500×.

**Figure 13 marinedrugs-23-00071-f013:**
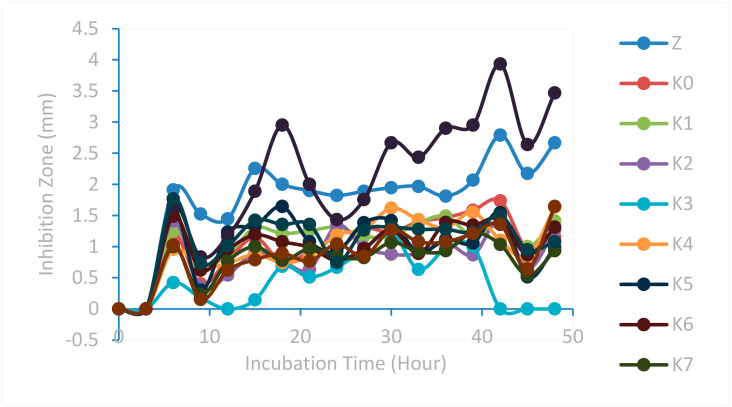
Graph of the relationship between bacterial inhibition zone diameter and incubation time against *Pseudomonas aeruginosa*.

**Figure 14 marinedrugs-23-00071-f014:**
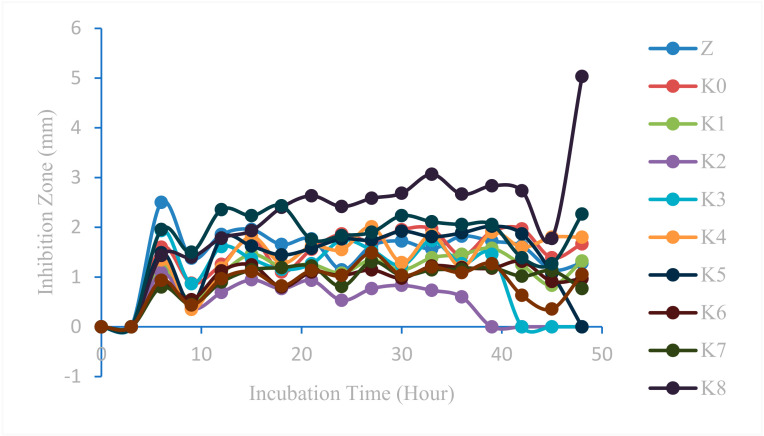
Graph of the relationship between inhibition zone diameter and incubation time against *Staphylococcus epidermidis*.

**Figure 15 marinedrugs-23-00071-f015:**
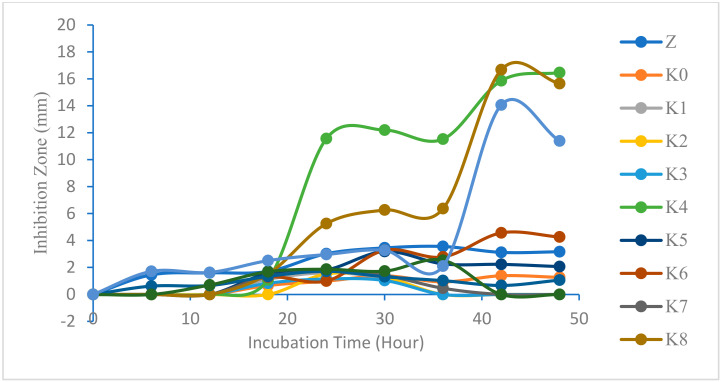
Relationship between the diameter of the inhibition zone and incubation time against *Malassezia furfur*.

**Table 1 marinedrugs-23-00071-t001:** Polydispersity index and particle size of ZnO nanoparticles.

Polydispersity Index	Particle Size (nm)	Vol%
0.537	1.74	1.4
	24.54	3.3
	164	20.5
	706	66
	3.860	4.1
	5.710	4.7

**Table 2 marinedrugs-23-00071-t002:** Simplification of the Tukey HSD test results for *Malassezia furfur* on sample types.

Sample Type	Average Diameter of the Resistance Zone (mm) ± SD
	*Malassezia furfur*
ZnO NPs	2.64 ± 0.887 ^a^
K0	1.10 ± 0.287 ^a^
K1	1.31 ± 0.255 ^a^
K2	1.39 ± 0.177 ^a^
K3	1.00 ± 0.160 ^a^
K4	11.45 ± 5.529 ^c^
K5	2.19 ± 0.558 ^a^
K6	2.86 ± 1.486 ^ab^
K7	1.22 ± 0.539 ^a^
K8	8.62 ± 6.109 ^bc^
K9	1.05 ± 0.395 ^a^
K10	1.70 ± 0.656 ^a^
K+	4.96 ± 4.883 ^ab^
K−	0 ± 0

Description: The average value of the diameter of the inhibition zone followed by different superscript letters shows significantly different results (significant), while the same superscript letters show significantly different results (insignificant) with a significance level of 95%.

## Data Availability

The data presented in this study are available on request from the corresponding author.

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
