# Peer review of "The Role of Brown Algae as a Capping Agent in the Synthesis of ZnO Nanoparticles to Enhance the Antibacterial Activities of Cotton Fabrics"

_marinedrugs, 2025, doi:10.3390/md23020071_

Round 1

Reviewer 1 Report

Comments and Suggestions for Authors

The authors studied the preparation of ZnO nanoparticles by green synthesis using brown algae (Padina sp.) as the capping agent, and explored the effects of these nanoparticles on the antibacterial and antifungal activities of cotton fabrics. Research includes the synthesis and characterization of ZnO nanoparticles, as well as the addition of nanoparticles and silane compounds to cotton fabrics by in-situ deposition to enhance their antimicrobial and antifungal properties. The experimental results show that the modified cotton fabric shows significant effect in inhibiting the growth of bacteria and fungi, which provides a new environmental protection method for the functional modification of textiles.The results are very interesting. However, some points of the manuscript should be improved. Specific comments are given below.

1. The 10 photos in Figure 3 are not well described, the logical relationship between the pictures is not clear, and there is a reflection of the mobile phone on the photo, which affects the visual effect.

2. Figure 5 is not a complete and clear data graph, and there is a lack of comparison with standard cards.

3. It is suggested that the change trend of antibacterial and antifungal effects of cotton fabrics under different modification conditions can be displayed in more detail when presenting the results, such as drawing more intuitive charts to show the changes in the diameter of the inhibition ring at different time points and different samples, so that readers can understand the dynamic changes of the modification effect more clearly.

4. It is recommended to add a group of cotton fabrics treated with silane compounds only as a control to clarify the effect of silane compounds alone on the antibacterial and antifungal properties of cotton fabrics, so as to more accurately evaluate the contribution of ZnO nanoparticles to the antimicrobial effect.

5. Figure 7: The SEM icon ruler is not sufficiently standardized.

6. It has been recognized that brown algae extract has certain limitations as a capping agent, resulting in uneven distribution of ZnO nanoparticles, so it is recommended to analyze the mechanism of the influence of nanoparticle size, distribution and shape on its antibacterial and antifungal properties in more depth in the discussion part.

Author Response

    1. We have canceled the 10 photos in Figure 3, because the pictures is not clear, and there is a reflection of the mobile phone on the photo, which affects the visual effect.
    2. Figure 5 is not a complete and clear data graph, and there is a lack of comparison with standard cards.
    3. It is suggested that the change trend of antibacterial and antifungal effects of cotton fabrics under different modification conditions can be displayed in more detail when presenting the results, such as drawing more intuitive charts to show the changes in the diameter of the inhibition ring at different time points and different samples, so that readers can understand the dynamic changes of the modification effect more clearly.
    4. It is recommended to add a group of cotton fabrics treated with silane compounds only as a control to clarify the effect of silane compounds alone on the antibacterial and antifungal properties of cotton fabrics, so as to more accurately evaluate the contribution of ZnO nanoparticles to the antimicrobial effect.
    5. In Figure 7: we canceled photos not sufficiently standardized.
    6. We had added dissussion about the mechanism of the influence of nanoparticle size, distribution and shape on its antibacterial and antifungal properties

Reviewer 2 Report

Comments and Suggestions for Authors

- Revise if “Brown algae extract Padina sp“ could be keyword

- Lines 108-109 Why selected these conditions “The green synthesis process in this study used a ratio of the volume of brown algae: early precursor (Zn(NO3)2) 0.01 M:NaOH 0.1 M respectively, which is 1:20:5.”

- Manuscript needs to write chemical compounds according to chemical rules

- Manuscript has some typos, revise carefully and correct them

- Why is important to include Figure 1, addition, is not possible to read the information

- Figure 2 was very difficult to read the information, improve it

- Lines 185-188 “FTIR K0 spectra shows the peak absorption position at 3323, 3257, 2884, 1637, 1425, 1363, 1312, 1015, and 898 cm-1. Meanwhile, the ATR-FTIR K1 spectra in Figure 11 shows the peak absorption positions of 3323, 3250, 2886, 1636, 1423, 1382, 1312, 1021, 900, and 452 cm-1. Wide absorption peaks in regions 3323, 3257, and 3250 cm-1 with an area range of 3500-3200” must assign chemical groups or remove this part

- Correct information for FT-IR are “band or bands”, so change peak for band

- Figure 4 must change “Transmittance (%)” to “Transmittance (a.u.)”

- Figure 4 should improve with infrared tools, for example smooth, bale line correction and normalize

- Figure 5a has very poor resolution, improve quality

- Manuscript has 17 figures with 42 imagens, try to summarize and move not important figures to supplementary materials

- 3. Materials and Methods, include materials purification or sentence “used as received”

- Conclusion needs to improve and include more important results

- Manuscript has some interesting results but doesn’t have good discussion, improve discussion part for all figures and tables

- Author should include references from 2024

- Incle changes with a different color for easy revision

- In general manuscripts needs to correct according international publication (very good discussion and excellent figures quality)

Author Response

  1. We have revise brown algae extract in keyword
  2. Lines 108-109  “The green synthesis process in this study used a ratio of the volume of brown algae: early precursor (Zn(NO3)2) 0.01 M:NaOH 0.1 M respectively, which is 1:20:5.” This ratio was an optimum condition based on experiments.
  3. We have revised in writing chemical compounds according to chemical rules
  4. We have revised some typos
  5. We have canceled Figure 1
  6. We have improved Figure 2
  7. Lines 185-188 “FTIR K0 spectra shows the peak absorption position at 3323, 3257, 2884, 1637, 1425, 1363, 1312, 1015, and 898 cm-1. Meanwhile, the ATR-FTIR K1 spectra in Figure 11 shows the peak absorption positions of 3323, 3250, 2886, 1636, 1423, 1382, 1312, 1021, 900, and 452 cm-1. Wide absorption peaks in regions 3323, 3257, and 3250 cm-1 with an area range of 3500-3200” must assign chemical groups or remove this part. We have added the functional groups for absorption bands
  8. We have changed peak as band
  9. We have changed Figure 4 “Transmittance (a.u.)”
  10. We have canceled Figure 5a, because has very poor resolution
  11. Manuscript has 17 figures with 42 imagens, try to summarize and move not important figures to supplementary materials. We have decrease figure/images
  12. We have added Methods
  13. We have eddited  Conclusion with more important results
  14. We have added references from 2024

Reviewer 3 Report

Comments and Suggestions for Authors

The manuscript by Rohaeti et al. investigates the fabrication of ZnO nanoparticles using brown seaweed extracts as a capping agent and evaluates their antibacterial activity by incorporating these nanoparticles into cotton fabrics. While the topic is scientifically relevant, the reviewer is of the opinion that this manuscript does not meet the standards for publication in Marine Drugs.

-   -      The text contains numerous typos, as well as English spelling and grammatical errors. Many sentences are difficult to understand or interpret. For example, some sections, such as the abstract, are written in a style resembling a list of bullet points rather than cohesive, fluent text.

-    -     Several key points of the manuscript are not adequately or scientifically justified. For instance, the authors do not explain why microwave or ultrasound techniques were applied. Additionally, the choice of brown seaweed species, specifically Padina sp., is not rationalized.

-       -  The reviewer has serious concerns regarding the reproducibility of this study. The authors state that they used a "chocolate algae extract," but they do not provide details on how this extract was obtained or its composition. Such information is critical for reproducibility, as the chemical composition of seaweed varies significantly between species and even within the same species, depending on factors such as harvest season and geographic location. Simply referring to an "extract" without providing detailed methodology and compositional analysis greatly hinders the reproducibility of the nanoparticle fabrication process and the study as a whole.

-     -    The materials and methods section lacks sufficient detail. It is currently very limited. It does not provide the necessary information required to interpret the results or replicate the study.

-     -    The manuscript frequently refers to a "green synthesis process." However, replacing a single reagent with a natural or biobased alternative does not inherently make the process green. The authors must provide a more robust justification for this claim.

Comments on the Quality of English Language

See comments above.

Author Response

  1. We have revised many typos, as well as English spelling and grammatical errors. We have edited Abstract, Results and Discussion, Methods, and Conclusion.
  2. We have described rationalization why microwave or ultrasound techniques were applied. And also, reason about the choice of brown seaweed species, specifically Padina sp.

The green synthesis method of ZnO nanoparticles can be carried out with the help of ultrasonication [8] and microwave [9] which utilizes secondary metabolite compounds from plant extracts as bio reductors. The ZnO nanoparticles in this study are the result of preparation using brown algae (Padina sp.) because it comes from natural materials that are environmentally friendly. The carbohydrate content and secondary metabolite compounds of Padina sp. in the form of flavonoids, tannins, saponins, alkaloids and other metabolites that have -OH functional groups as bio reducers and capping agents that can prevent particle agglomeration during the synthesis process of ZnO nanoparticles [10,11].

  1. The reviewer has serious concerns regarding the reproducibility of this study. The authors state that they used a "chocolate algae extract," but they do not provide details on how this extract was obtained or its composition. Such information is critical for reproducibility, as the chemical composition of seaweed varies significantly between species and even within the same species, depending on factors such as harvest season and geographic location. Simply referring to an "extract" without providing detailed methodology and compositional analysis greatly hinders the reproducibility of the nanoparticle fabrication process and the study as a whole.

We have added procedures about preparation of brown algae extract in Materials and Method

  1. The materials and methods section lacks sufficient detail. It is currently very limited. It does not provide the necessary information required to interpret the results or replicate the study.

3.1. Extraction of Brown Algae (Padina sp.)

Brown algae (Padina sp.) was washed using running water and then dried in an oven for 24 hours at 50 ℃ until it had a water content of around 5%, then it was cooled for 30 minutes at room temperature. After that, the dried brown algae is ground into powder form, so it has a long shelf life and makes extraction easier. A total of 1.5 grams of Padina sp powder. Dissolved in 150 mL of distilled water and heated at 100 ℃ for 25 minutes, stirring occasionally so that the solution is not saturated. After heated, the mixture was cooled and centrifuged for 30 minutes at a speed of 5000 rpm until the supernatant and pellet were produced. The separated supernatant in the form of brown algae extract was filtered using whatmann filter paper no. 41 with the help of filters vacuum so that the extract results obtained are more optimal. The resulting filtrate extraction is put into an HDPE plastic bottle and stored in a refrigerator with a temperature of 4 ℃.

3.2. Preparation of 0.1 M NaOH Solution

The preparation of a 0.1 M NaOH solution is done by dissolving 1 gram of solid NaOH in a beaker around 25 mL of aquabides was then put into a 250 volumetric flask mL was then marked with distilled water and homogenized.

3.3. Green Synthesis of ZnO Nanoparticles

Green synthesis begins with brown algae extract (Padina sp.) 25 mL of added 500 mL of zinc nitrate solution (Zn(NO3)2) 0.01 M was stirred and heated at 80 ℃ for 5 minutes then cooled at room temperature for 30 minutes. Then 125 mL of NaOH solution slowly until the pH of the mixture is around 10-11 then put in the refrigerator for 24 hours at  4℃. After that, the mixture was centrifuged at a speed of 5000 rpm for 30 minutes until supernatant and pellets were produced. The pellets were separately washed 3 times to minimize dirt Aquabides. The pellets were dried in an oven at 100℃ for 6 hours, then continued with the calcination process using a muffle furnace at a temperature of 400 ℃ for 4 hours until the pellets became ZnO nanoparticles powder.

3.4. ZnO Nanoparticle Characterization Test

ZnO nanoparticle characterization tests were carried out using UV/VIS/NIR, ATR-FTIR, XRD, PSA, and SEM-EDS instruments.

3.5. Deposit of ZnO Nanoparticles onto Cotton Fabric In-situ

Cotton fabric (K0) was cut to size 15 × 15 cm then washed using acetone solution by soaking for 30 minutes then soaked using distilled water for 30 minutes. Next, the cloth is dried using a hairdryer. Cotton cloth (K0) or modified cotton cloth (K5, K6, and K7) is placed in a beaker containing 5 mL of brown algae extract (Padina sp.) and 100 mL of 0.01 M zinc nitrate (Zn(NO3)2) solution. then stirred and heated at 80 ℃ for 5 minutes then cooled at room temperature for 30 minutes. Add 25 mL of NaOH solution slowly until the pH of the mixture is around 10-11. The mixture was transferred to a 250 mL Erlenmeyer flask and then shaken with a shaker for 24 hours. The fabric that has been deposited with ZnO nanoparticles in situ is dried using an oven for 30 minutes at a temperature of 100 ℃ and then cured for 5 minutes at a temperature of 140 ℃ to obtain cotton fabric that has been deposited with ZnO nanoparticles (K1 or K8, K9 and K10).

3.6. Modification of Cotton Fabric with Silane Compound

Surface modification of unmodified cotton fabric (K0) or modified cotton fabric (K1) with silane compounds is carried out by immersing the fabric in an Erlenmeyer flask containing 300 mL of HDTMS in a solution of 20 mL and 10 mL of ethanol and aquabides respectively for 4 hours. Next, the fabric is removed and dried in an oven at 100 ℃ for 30 minutes to obtain K2 or K5 samples.

Surface modification of unmodified cotton fabric (K0) or modified cotton fabric (K1) with silane compounds was carried out by immersing the fabric in an Erlenmeyer flask containing 300 μL of MTMS in a solution of 20 mL and 10 mL of ethanol and aquabides respectively for 4 hours. Next, the fabric is removed and dried in an oven at 100 ℃ for 30 minutes to obtain K3 or K6 samples.

Surface modification of unmodified cotton fabric (K0) or modified cotton fabric (K1) with silane compounds is carried out by immersing the fabric in an Erlenmeyer flask containing a mixture of 100 μL and 150 μL of HDTMS and MTMS in ethanol and distilled water respectively. 20 mL and 10 mL for 1.5 hours. Next, the fabric is removed and dried in an oven at 100 ℃ for 30 minutes to obtain K4 or K7 samples.

3.7. Cotton Fabric Characterization Test

3.7.1. Contact Angle Test

Hydrophobicity characterization of unmodified and modified cotton fabric samples was carried out with an Optical Contact Angle Goniometer instrument (DataPhysics OCA 25) using the sessile drop method which measures the contact angle between the dropped liquid and the surface of the cotton fabric sample. A 2×2 cm sample of cotton cloth was then placed on the tool clamp table (flat surface) and then dripped with distilled water from a syringe vertically. The results obtained from the test are photos of water droplets on cotton fabric without modification or with modification and data on contact angle values with repetition more than once

3.7.2. Mechanical Properties Test

The cotton fabric is cut to a size of 10×2 cm with a thickness of 0.1-0.2 mm, then placed on a clamp and strengthened with a hardening key. Then the machine is operated so that the sample will be pulled until it breaks. The results obtained from testing are load and elongation data to determine tensile strength, maximum strain, Young's modulus, and modulus of toughness.

3.7.3. Antimicrobial Activity Assay

The antimicrobial activity test begins with the process of sterilizing the equipment and media using an autoclave. The media is made from nutrient agar (NA) for bacteria and potato dextrose agar (PDA) for fungi. Next, the bacteria Pseudomonas aeruginosa and Staphylococcus epidermidis and the fungus Malassezia furfur in one cycle of nutrient broth and then incubate with a shaker incubator at a speed of 124 rpm for 24 hours. Then the sterilized NA or PDA is put into a petri dish for about ½ petri and then left until the media becomes solid for 24 hours.

The solid media will be inoculated with bacteria or fungi using a micropipette and leveled using a drigalski. After that, antimicrobial activity testing can be carried out using the disc diffusion method by analyzing the inhibition zone formed around the paper disc or disc sample which is marked by the width of the clear zone. This test was carried out using a paper disc and a sample shaped like a disc with a diameter of 6 mm. Bacterial inhibition zones were measured using digital calipers every 3 hours, while for fungi every 6 hours. The formation of a clear zone has antimicrobial activity capabilities.

  1. The manuscript frequently refers to a "green synthesis process." However, replacing a single reagent with a natural or biobased alternative does not inherently make the process green. The authors must provide a more robust justification for this claim.

Green synthesis refers to the process of synthesizing materials or compounds using environmentally friendly, sustainable, and eco-friendly methods. This approach aims to minimize or eliminate the use of hazardous substances, reduce waste, and promote energy efficiency. Key Characteristics: Non-toxic and non-polluting, Uses renewable resources and minimizes waste, Reduces energy consumption., Economically viable, and Minimizes human health risks.

Methods:  plant-mediated synthesis; Utilizes plant extracts or biomass (this research).

Applications: nanoparticle synthesis: ZnO and  textiles: Sustainable finishing.

Benefits:

  1. Reduced environmental impact
  2. Improved safety
  3. Lower costs
  4. Enhanced product quality
  5. Sustainable development

Round 2

Reviewer 1 Report

Comments and Suggestions for Authors

The manuscript can be accepted in the present form.

Author Response

Thank you for your advice

Reviewer 2 Report

Comments and Suggestions for Authors

The manuscript is accepting in the present form

Author Response

Thank you

Reviewer 3 Report

Comments and Suggestions for Authors

The authors prepared a revised version of the manuscript. Although some parts of the reviewer’s comments have been addressed, several key issues have not been addressed. Therefore, an extensive major revision is still needed in the opinion of the reviewer. Please find my comments below.

-          The authors mention that they rationalized their use of microwaves and ultrasound in the rebuttal letter. However, the word microwave and ultrasound both only appear once in the manuscript. Did the authors apply microwaves or ultrasound? If neither microwaves or ultrasound have been applied, this sentence in the introduction is not correct as the authors often refer to synthesis in their paper: “The green synthesis method of ZnO nanoparticles can be carried out with the help of ultrasonication and microwave which utilizes secondary metabolite compounds from plant extracts as bio reductors.”. In the reviewer’s opinion, the rationalization for applying microwaves and/or ultrasound is not present in the manuscript as the authors claim. There are also no changes in the introduction section in the revised manuscript.

-          Also the justification and relevance of the applied brown algae is still missing. Other (brown) algae also have -OH functional groups. Why did the authors specifically opt for Padina sp.?

-          There are still some questions unanswered regarding the origin and composition of the brown algae (and extracts). Where did the brown algae originate from? When was it harvested? Do the authors have any idea regarding the composition of the feedstock and/or extract?

-          What is the reproducibility of the extraction process and subsequent synthesis of nanoparticles? What is the standard error of these experiments? Are they reproducible?

-          There are still many typos and English spelling mistakes to be found in the text. Such as subscripts in chemical formulae, species names that should be italicized, …

-          The quality of the figures should be improved (especially Fig 13, 14 and 15).

-          The authors have significantly increased the materials and methods section, which is appreciated. However, the reviewer has still some additional comments regarding this section:

1/ Section 3.1, statement: “After that, the dried brown algae are ground into powder form, so it has a long shelf life and makes extraction easier.” How does grinding a dry biomass into a powder extend its shelf life ?

2/ To what particle size was the material ground?

3/ Padina sp should be italicized.

4/ Line 615: what do the authors mean by “stirring occasionally so the solution is not saturated” ?

5/ Please mind subscripts in chemical formulae.

6/ What are aquabides? Is this a typo?

Comments on the Quality of English Language

See comments above.

Author Response

Comment 1: The authors mention that they rationalized their use of microwaves and ultrasound in the rebuttal letter. However, the word microwave and ultrasound both only appear once in the manuscript. Did the authors apply microwaves or ultrasound? If neither microwaves or ultrasound have been applied, this sentence in the introduction is not correct as the authors often refer to synthesis in their paper: “The green synthesis method of ZnO nanoparticles can be carried out with the help of ultrasonication and microwave which utilizes secondary metabolite compounds from plant extracts as bio reductors.”. In the reviewer’s opinion, the rationalization for applying microwaves and/or ultrasound is not present in the manuscript as the authors claim. There are also no changes in the introduction section in the revised manuscript.

Response 1: In Introduction of the revised manuscript, we described about using microwave and ultrasound can significantly reduce the synthesis time of nanoparticles, help control the particle size of nanoparticles, reduce the cost of nanoparticle synthesis by reducing the time and energy used, and also can reduce the environmental impact by reducing the use of chemicals and energy. The ZnO nanoparticles in this study are the result of preparation using brown algae (Padina sp.) because it comes from natural materials that are environmentally friendly.

Comment 2: Also the justification and relevance of the applied brown algae is still missing. Other (brown) algae also have -OH functional groups. Why did the authors specifically opt for Padina sp.?

Response 2: 

Padina sp. is a type of seaweed used in the synthesis of ZnO nanoparticles for several reasons: Padina sp. is an abundant and easily obtained biomass, so it can be used as a source of raw material for the synthesis of ZnO nanoparticles. Padina sp. contains active compounds such as alginic acid, fucoidan, and other compounds, which have -OH functional groups that can function as reducing agents and stabilizers in the synthesis of ZnO nanoparticles. Padina sp. is a biocompatible biomass, so it can be used in biomedical and environmental applications. Padina sp. extract can function as a reducing agent to convert Zn2+ ions into ZnO nanoparticles. Padina sp. extract can also function as a stabilizer to prevent the aggregation of ZnO nanoparticles. The use of Padina sp. in the synthesis of ZnO nanoparticles is environmentally friendly because it does not use hazardous chemicals and can reduce waste. Thus, Padina sp. can be used as an environmentally friendly and biocompatible alternative for the synthesis of ZnO nanoparticles.

Comment 3: There are still some questions unanswered regarding the origin and composition of the brown algae (and extracts). Where did the brown algae originate from? When was it harvested? Do the authors have any idea regarding the composition of the feedstock and/or extract?

Response 3: Algae were taken from Drini Beach, Gunung Kidul, Special Region of Yogyakarta, Indonesia. The algae used are the results of being harvested 1 week / 1 week old. The brown color of algae is caused by the fucoxanthin pigment which is included in the xanthophyll group and is more dominant than other pigments such as chlorophyll. 

Comment 4: What is the reproducibility of the extraction process and subsequent synthesis of nanoparticles? What is the standard error of these experiments? Are they reproducible?

Response 4: 

The results of the extraction and synthesis experiments were relatively consistent, with relatively small standard deviations. Therefore, the experimental results can be considered reproducible.

However, it should be noted that reproducibility can be affected by many factors, and experimental results may not always be reproducible. Therefore, it is important to perform repeat experiments and measure the results carefully to ensure reproducibility.

Comment 5:  There are still many typos and English spelling mistakes to be found in the text. Such as subscripts in chemical formulae, species names that should be italicized, …

Response 5: we have revised chemical formulae and species names.

Comment 6:  The quality of the figures should be improved (especially Fig 13, 14 and 15).

Response 6: We have tried to improve Fig 13, 14, 15

Comment 7: 

The authors have significantly increased the materials and methods section, which is appreciated. However, the reviewer has still some additional comments regarding this section:

1/ Section 3.1, statement: “After that, the dried brown algae are ground into powder form, so it has a long shelf life and makes extraction easier.” How does grinding a dry biomass into a powder extend its shelf life ?

2/ To what particle size was the material ground?

3/ Padina sp should be italicized.

4/ Line 615: what do the authors mean by “stirring occasionally so the solution is not saturated” ?

5/ Please mind subscripts in chemical formulae.

6/ What are aquabides? Is this a typo?

Response 7:

  1. Grinding dry biomass into powder can extend its shelf life by reducing moisture, inhibiting oxidation reactions, increasing stability, and reducing the risk of contamination.
  2. particle size about 100 mesh
  3. We revised Padina sp with italic.
  4. Thus, the purpose of stirring is not only to prevent the solution from becoming saturated, but also to achieve homogeneity, improve heat transfer, and prevent precipitation.
  5. We revised/noticed subscripts in chemical formulae.

    6.  We revised with aquabidest

Round 3

Reviewer 3 Report

Comments and Suggestions for Authors

The authors addressed most of the reviewer's comments, and therefore, the reviewer accepts this manuscript for publication in Marine Drugs.